# Feasibility and Reliability of the Osteoarthritis Quality Indicator Questionnaire for Assessing Osteoarthritis Care in Bilingual General Practices in South Tyrol/Alto Adige, Italy

**DOI:** 10.3390/medicina61111921

**Published:** 2025-10-26

**Authors:** Christian J. Wiedermann, Antje van der Zee-Neuen, Pasqualina Marino, Angelika Mahlknecht, Sonja Wildburger, Julia Fuchs, Christian Dejaco, Michele di Lernia, Giuliano Piccoliori, Adolf Engl, Markus Ritter, Nina Østerås

**Affiliations:** 1Institute of General Practice and Public Health, Claudiana College of Health Professions, 39100 Bolzano, Italy; 2Center for Physiology, Pathophysiology and Biophysics, Institute of Physiology and Pathophysiology, Paracelsus Medical University, 5020 Salzburg, Austriamarkus.ritter@pmu.ac.at (M.R.); 3Gastein Research Institute, Paracelsus Medical University, 5020 Salzburg, Austria; 4Ludwig Boltzmann Institute for Arthritis and Rehabilitation, 5020 Salzburg, Austria; 5Department of Rheumatology, Hospital of Brunico (ASAA-SABES), Teaching Hospital of the Paracelsus Medical University, 39031 Brunico, Italy; 6Department of Rheumatology and Immunology, Medical University of Graz, 8010 Graz, Austria; 7School of Medical Sciences, Kathmandu University, Dhulikhel 45200, Nepal; 8Center for Treatment of Rheumatic and Musculoskeletal Diseases (REMEDY), Diakonhjemmet Hospital, 0319 Oslo, Norway

**Keywords:** primary care, questionnaire validation, health services research, patient-reported outcomes, musculoskeletal disorders

## Abstract

*Background and Objectives*: Evaluating osteoarthritis (OA) care quality is increasingly relevant for service improvement and benchmarking purposes. The Osteoarthritis Quality Indicator questionnaire (OA-QI) measures patient-reported guideline-concordant care; however, no version has been tested in Italian primary care or bilingual contexts. This study aimed to introduce the OA-QI version 3 (OA-QI v3) in German and Italian, assess its applicability in practice, and examine its acceptability and reliability. *Materials and Methods*: A cross-sectional survey was conducted using the South Tyrolean General Practice Research Network. Thirty-eight general practitioners recruited 266 patients with hip or knee OA. Patients completed the OA-QI v3 in German or Italian, with subsamples for comprehensibility testing (*n* = 38) and retest reliability after 14 days (*n* = 36). Test–retest reliability was analyzed using percent agreement, Cohen’s κ, intraclass correlation coefficients (ICC), and standard error of measurement. The smallest detectable change was analyzed to estimate factual change. *Results*: Response rate reached 95% of the targeted patients. Patient feedback showed good comprehensibility and ease of use in both languages. Adherence to recommended quality indicators varied, with strengths in physical activity advice, NSAID prescription, and pain assessment, but gaps in weight management, occupational counseling, and assistive devices. Test–retest reliability ranged from fair to substantial at the item level (κ = 0.33–0.69) and was moderate for the total score (ICC = 0.55, 95% CI 0.28–0.74). While measurement error restricted individual-level interpretation, reliability at the practice or institutional level supports application for benchmarking and quality monitoring. *Conclusions*: The OA-QI v3 was feasible, acceptable, and reliable for group-level assessments in South Tyrol. These findings position OA-QI v3 as a practical tool for identifying care gaps and guiding quality improvement, while providing important lessons for the full validation of the German and Italian versions in larger cross-national samples.

## 1. Introduction

Osteoarthritis (OA) of the hip and knee is one of the most prevalent chronic musculoskeletal disorders, affecting over 500 million individuals globally. Its prevalence is increasing in Europe and worldwide, primarily due to aging populations, obesity, and joint injuries [1,2,3]. Knee OA is the most common site and a leading contributor to years lived with disability, while hip OA also imposes a significant burden, with notable sex differences in prevalence across both conditions [4]. Collectively, hip and knee OA are major contributors to reduced physical function, impaired quality of life, and rising healthcare costs, posing a central challenge to health systems [5,6]. Given the demographic shift towards aging societies and the projected substantial increase in osteoarthritis prevalence over the coming decades [7,8,9], effectively addressing OA through prevention, early diagnosis, and evidence-based management strategies has become a pressing priority for clinical practice and public health.

Exercise and physical activity are fundamental evidence-based interventions for hip and knee OA, as substantiated by systematic reviews, meta-analyses, and international guidelines, which consistently demonstrate benefits in pain relief, functional improvement, and prognosis [10,11,12,13,14]. Interventions ranging from aerobic and strength-training programs to mind–body approaches and aquatic exercise yield effects comparable to pharmacological therapies, while also enhancing quality of life and aiding in the management of comorbidities such as cardiovascular disease and diabetes [12,13,15]. No single exercise modality has been proven superior, as all forms of regular activity appear to be beneficial, with moderate-intensity exercise performed 2–3 times weekly emerging as a pragmatic recommendation [14,16]. Exercise is generally safe for most patients with OA, with a minimal risk of serious adverse events [11,17]. Guidelines consistently advocate exercise, weight management, and self-management as first-line strategies [7,10]. However, despite this evidence and the central role of physical activity in OA management, implementation remains suboptimal. Contributing factors include patient barriers (such as pain, kinesiophobia, and low motivation), clinician factors (including outdated beliefs and limited confidence in prescribing exercise), and systemic issues (such as lack of access to programs and insufficient adherence support) [18,19,20].

The evaluation of care quality in chronic disease management is often structured according to conceptual frameworks such as Donabedian’s model, which distinguishes between structural, process, and outcome dimensions of care quality [21], and the Chronic Care Model, which emphasizes patient-centered organization of care and self-management support [22]. The OA-QI questionnaire primarily targets the process dimension, capturing patient-reported receipt of guideline-concordant care as a key determinant of overall care quality.

Standardized measurement tools are essential for assessing and improving the quality of OA care. The Osteoarthritis Quality Indicator (OA-QI) questionnaire provides structured, measurable items that translate guideline recommendations into auditable domains, such as assessment, education, exercise counseling, weight management, pharmacological management, and follow-up [23,24]. Developed through systematic reviews and expert consensus, often including patient involvement, OA-QI sets have demonstrated their validity and feasibility for use in both clinical audits and patient-reported surveys [25,26]. International adaptation across various countries consistently revealed the underuse of non-pharmacological strategies, particularly exercise and lifestyle counseling, highlighting their role in benchmarking and guiding quality improvement initiatives [27].

Despite significant international advancements in the development of OA quality indicators, there is a deficiency in validated, regionally adapted instruments for South Tyrol and Italy. The bilingual (i.e., German and Italian) healthcare context of the region has constrained the capacity of general practice networks to systematically measure quality and to benchmark outcomes. The current initiative addresses this issue by translating the OA-QI questionnaire into German and Italian for application within South Tyrol’s healthcare setting, thereby facilitating standardized patient-reported assessments of OA care [28]. This adaptation ensures cultural relevance and enables cross-border comparisons with future data from Austria and other German speaking countries. Up until now, the broader Italian primary care landscape lacked such standardized tools, and reviews have indicated deficiencies in OA management practices [29,30]. On this line, the South Tyrol project will establish a foundation for quality monitoring while also supporting international benchmarking and testing the feasibility of integrating the South Tyrolean system. Feasibility testing is a critical methodological step when introducing standardized instruments into new cultural and health system contexts, as it ensures that questionnaires are both linguistically adapted and practically applicable before broader validation [31,32,33]. Without this process, even well-validated instruments may fail to accurately capture the intended constructs, leading to invalid results [34,35]. Therefore, the present study evaluated the feasibility, acceptability, and test–retest reliability of the German and Italian versions of the OA-QI v3 questionnaire in general practice, while also providing an initial description of care gaps, particularly in patient education, exercise, and lifestyle counseling.

## 2. Materials and Methods

### 2.1. Study Design and Setting

This study was designed as a cross-sectional, practice-based survey within the South Tyrolean General Practice Research Network (SAMNET), a regional collaboration of general practitioners (GPs) engaged in health services research [36].

The bilingual context of South Tyrol provided the opportunity to test both language versions while considering the cultural and organizational features of primary care in Italy. The a priori targeted sample size was 280 patients, based on 40 GPs each enrolling seven consecutive eligible patients during routine consultation. This approach mirrored international OA-QI validation studies conducted in Norway [25] and the Netherlands [37], but was adapted to the South Tyrolean setting.

Sample size considerations were pragmatic and reflected the recruitment capacity of the network. For questionnaire validation, subject-to-item ratios of approximately 1:5 to 1:30 are considered sufficient [38]. With 17 OA-QI items, the targeted sample size corresponded to a ratio of approximately 1:16, which lies well within this recommended range and allows estimation of feasibility and acceptability proportions with acceptable precision (±6% at 95% confidence). For test–retest reliability, a minimum of 50 participants is recommended [39]; our planned retest sample of about 40, and the final achieved *n* = 36, therefore provided preliminary but informative estimates of measurement stability in this early-stage reliability assessment.

### 2.2. Participants

#### 2.2.1. General Practitioners

Forty GPs affiliated with the SAMNET were invited to participate. Each physician was asked to recruit seven consecutive patients with hip or knee OA during routine consultations. To ensure methodological completeness, each practice contributed to the main survey and specific subgroups: one patient per practice was assigned to the comprehensibility assessment, and two patients were assigned to the test–retest reliability assessment. The GPs were responsible for informing the patients, obtaining written consent, and assigning anonymized study codes. The completed questionnaires were sealed by the patients in envelopes and returned in bulk to the study center.

#### 2.2.2. Patients

Patients were eligible if they (i) had a known diagnosis of hip or knee OA or presented with knee or hip pain and fulfilled NICE-based clinical criteria (age ≥ 45 years, activity-related joint pain, and no or <30 min of morning stiffness) [40], and (ii) were able to complete the questionnaire in German or Italian language. The exclusion criteria were as follows: malignant disease, rheumatoid or other inflammatory arthritis, Kellgren–Lawrence grade IV joint degeneration, other inflammatory rheumatic conditions, psychiatric or cognitive disorders preventing participation, and concurrent involvement in other clinical studies.

#### 2.2.3. Recruitment Flow

Of the 40 invited GPs, 38 actively participated (95.0%). Together, they recruited 266 patients, corresponding to 95.0% of the planned sample size of 280. All patients completed the main OA QI questionnaire. In addition, 89 patients completed the retest questionnaire approximately 14 days later, and 40 patients completed the comprehensibility module.

### 2.3. Study Registration

This feasibility and reliability study was conducted within the framework of a project previously described in a published study protocol [28]. This protocol outlines the stepwise translation, cultural adaptation, and validation of the OA-QI questionnaire [25] for use in Germany and Italy. Although the feasibility phase reported here was not separately registered, it represents the first step of a larger program of research that is registered in the ISRCTN registry (ISRCTN93874734) and includes a planned combined South Tyrol–Salzburg cohort.

### 2.4. Instruments

#### 2.4.1. Osteoarthritis Quality Indicator Questionnaire (OA-QI v3)

The OA-QI questionnaire was originally developed and validated in Norway [25] and has since been applied internationally to evaluate the implementation of guideline-recommended care for hip and knee OA [37,41]. The most recent version, OA-QI v3, was developed under the leadership of Nina Østerås, together with an international working group, including two of the authors of the current study (A. Z.-N. and C. J. W.). The English OA-QI v3 builds on OA-QI v2 [25] by broadening the context of information sources, refining the wording of several questions (notably weight advice, work, non-steroidal anti-inflammatory drugs (NSAIDs), and side effect-related questions), and restructuring the numbering of items. OA-QI v3 specified the question regarding medication uptake by asking if NSAIDs were the first medication recommended and listing examples, while the side effects question is reframed in terms of actual use. Version 3 also clarifies the scope of work (paid or unpaid) and introduces slight wording changes for improved clarity. Overall, OA-QI v3 preserves the original 17 core indicators but modernizes terminology and aligns the pharmacologic item with current international guidelines on OA management. A detailed item-by-item comparison of OA-QI v2 and OA-QI v3, including the nature and rationale of each modification, is provided in Appendix A. The OA-QI v3 has not yet been formally published.

The questionnaire covered the following domains of OA management:Information and counseling: provision of information about OA and treatment options; advice and instruction on exercise; counseling on coping and daily management; and advice and support for weight reduction (if overweight).Assessment: evaluation of pain, activity limitations, and need for walking aids; discussion of occupational aspects.Follow-up and pharmacological treatment: control visits; prescription of NSAIDs and information on side effects; and corticosteroid injections and information on side effects.Surgical treatment: Information about and/or referral for joint replacement surgery.

Each item has three response options: (i) Yes (the indicator was achieved). (ii) No (the indicator was not achieved). (iii) Not relevant/do not remember (the indicator was not applicable).

Following established scoring procedures, items marked “not relevant/do not remember” were treated as “not eligible” and excluded from the denominator of the percentage calculation. For each participant, an individual OA-QI achievement score was calculated (Equation (1)).(1)$$OA\text{-}QI\;score\;(\%)=\frac{Number\;of\;“Yes”\;responses}{Number\;of\;eligible\;items\;(Yes+No)}\times 100$$

Patients with fewer than four eligible items were excluded from the analysis to avoid unstable estimates. At the item level, achievement rates were expressed as the percentage of eligible patients responding “Yes.” At the patient level, achievement was summarized using the mean, standard deviation, median, and interquartile range. Distributions were checked for floor and ceiling effects, defined as >15% of patients scoring 0% or 100%. For comparability, the analytic procedures followed those reported in the validation study from Norway [25] and the application in the Netherlands [37].

##### OA-QI v3 Translation and Adaptation

The German (OA-QI v2-D) and Italian (OA-QI v2-I) versions of the OA-QI were initially produced by a professional translation agency (Wilkens Translation Agency, Leiden, Netherlands) specializing in medical translations and certified according to ISO 9001, ISO 17100, and ISO 13485. For the present study, these versions were adapted to the updated OA-QI v3, and translated from English into German and Italian by native-speaking translators. No back-translation was performed because the main work was performed by a certified medical translation provider. Subsequently, both language versions (OA-QI v3-D and OA-QI v3-I) were reviewed by two rheumatologists who were native speakers of German (C.D.) and Italian (M.L.). This step ensured cultural appropriateness and patient comprehensibility, with terms adapted for clarity, such as rendering “NSAIDs” as “schmerzstillende Entzündungshemmer” in German and “farmaci antidolorifici e antiinfiammatori” in Italian (Appendix A).

#### 2.4.2. Additional Variables

In addition to the 17 OA-QI items, the primary questionnaire gathered data on the clinical characteristics of OA and the sociodemographic factors of the participants.

Among the clinical variables, patients reported whether they had ever received a formal diagnosis of OA and indicated the frequency of joint pain (daily, weekly, rarely, or never). They were asked to specify the joints most affected (left or right hip, left or right knee, or other joints) and whether their pain impaired daily activities, with response options ranging from “no” to “constantly.” Further items assessed the use of physiotherapy or exercise therapy (yes/no) and whether the patients had undertaken lifestyle changes due to OA (no, more physical activity, dietary changes, or both). The questionnaire also inquired about the presence of comorbidities (no/yes, with specification) and the duration of symptoms, with response categories ranging from less than 1 year to >20 years. A detailed history of joint replacement surgery was collected, including recommendations or operations performed on the hip or knee, with the year of surgery, if applicable. Finally, patients were asked about their sources of information on OA and its treatment, with options including GPs, orthopedic surgeons, rheumatologists, physiotherapists, nurses, pharmacists, other patients, family or friends, and the Internet or social media.

Sociodemographic information included year of birth, sex (male, female, diverse, or prefer not to answer), native language (German, Italian, Ladin, or other), educational attainment (compulsory school, vocational training, high school, or university degree), and employment status (full-time, part-time, retired, unemployed, or other). Additionally, patients were asked to rate their subjective economic situation on a five-point scale ranging from “very good” to “very difficult.”

### 2.5. Comprehensibility and Acceptability Questionnaire

To evaluate comprehensibility and acceptability, a stand-alone questionnaire was administered to a predefined subsample of patients, with one participant per practice site. This instrument included items assessing the overall clarity of the OA-QI questions, ranging from “very well understandable” to “not understandable,” the presence of questions perceived as difficult, and the identification of unclear terms. Additional items addressed the ease of completing the questionnaire, from “very easy” to “very difficult,” the perceived time burden, and the comprehensibility of the response options. Furthermore, patients were invited to provide open-ended feedback on the wording and potential improvements.

Patients were selected according to the study protocol, with one patient per participating practice invited to complete a comprehensibility module. German- and Italian-speaking patients were included to ensure the representation of the bilingual study setting. Analyses were conducted overall and stratified by language subgroups. Responses to the four-point and dichotomous scales were summarized descriptively, and open-text feedback was reviewed qualitatively to identify recurring issues and suggestions for improvement.

### 2.6. Retest for Reliability

Test–retest reliability was assessed in a predefined subsample of patients, with two participants per practice invited to complete the OA-QI questionnaire again after approximately 14 days. The retest questionnaire repeated the 17 OA-QI items and included four additional questions on consultations with healthcare professionals, interim changes in osteoarthritis symptoms, new or modified medication, and other treatments initiated since the first administration. Participant responses were excluded from the test–retest analyses if they had visited a general practitioner, medical specialist, or physiotherapist during the interim period. Patients were identified by a combined practice and patient ID, allowing linkage between baseline and retest responses.

### 2.7. Statistical Analyses

All analyses were performed using IBM SPSS Statistics (version 27.0; IBM Corp., Armonk, NY, USA). Data were screened for completeness and plausibility; patients with missing responses on all OA-QI items or fewer than four eligible items were excluded from the study. Descriptive statistics (frequencies, percentages, means with standard deviations (SD), medians with interquartile ranges (IQR) were used to summarize the sample characteristics, OA-QI item responses, and overall achievement scores. Floor and ceiling effects were defined as >15% of patients scoring 0% or 100% on the patient-reported outcome measure, respectively. Bivariate associations between OA-QI achievement scores and patient characteristics were examined using correlation analyses (Spearman’s rank correlation coefficient for ordinal and categorical variables, and Pearson’s correlation coefficient for continuous age). Effect sizes were interpreted according to Cohen’s thresholds.

The test–retest reliability of OA-QI v3 was assessed in a predefined stable subsample. Item-level agreement was quantified using percent agreement and Cohen’s kappa (κ) with 95% confidence intervals, as interpreted by Landis and Koch [42]. For the overall achievement score, intraclass correlation coefficients (ICC; two-way mixed effects, absolute agreement) were calculated. Measurement error was further characterized using the standard error of measurement (SEM) and the smallest detectable change (SDC) at the individual and group levels, derived from ICC and the pooled SD.

Multivariable mixed-effects linear regression models were fitted with the OA-QI achievement score (%) as the dependent variable to assess independent associations with sociodemographic, clinical, and contextual characteristics. All variables showing significant bivariate associations were considered for inclusion. The number and use of information sources were excluded from the final model to maintain conceptual coherence, as it may represent an intermediate behavioral factor rather than a structural determinant of care quality. Predictors included age (continuous), sex (female vs. male), education (ordinal, per higher category), economic status (ordinal, higher values indicating worse status), comorbidity (≥1 chronic disease vs. none), prior joint surgery (yes/no), number of affected joints (0–4), pain frequency, activity limitation, duration of symptoms, language group (Italian, Ladin, or other vs. German as the reference), and number of information sources. A random intercept for general practitioners accounted for clustering within practices. Since each patient contributed only one observation, no random effect for patient ID was necessary. Models were estimated using restricted maximum likelihood (REML) with robust standard errors. The intraclass correlation coefficient (ICC) was calculated to quantify the proportion of variance in OA-QI scores attributable to practice-level clustering. Multicollinearity was assessed using variance inflation factors (VIF), and model fit was evaluated using −2 log likelihood (−2LL), Akaike’s Information Criterion (AIC), and Bayesian Information Criterion (BIC).

All statistical tests were two-sided, and *p*-values < 0.05 were considered statistically significant.

### 2.8. Use of Generative Artificial Intelligence Tools

Generative AI tools, specifically ChatGPT-5 (OpenAI), were used to support the structuring and linguistic refinement of the manuscript, including improved clarity and consistency in the Methods sections, particularly those describing the translation and adaptation process (Section 2.3) and statistical analysis (Section 2.5). AI assistance was used to generate and refine SPSS syntax for data preparation and analysis. All statistical computations and analyses were then conducted and verified by the authors. No AI tools were used for the execution of data analysis, statistical computation, or interpretation of results. The authors have reviewed and edited the output and take full responsibility for the content of this publication.

## 3. Results

### 3.1. Study Population

A total of 40 GPs from the SAMNET were invited to participate in the study. Of these, 38 practices actively recruited patients, while 2 did not contribute, resulting in an overall GP participation rate of 95.0%. A total of 266 patients with hip or knee osteoarthritis were enrolled, corresponding to 95.0% of the planned sample size of 280 participants. All included patients completed the OA-QI questionnaire, with subsamples further completing the comprehensibility module (*n* = 38) and test–retest reliability assessment (*n* = 73). Among the 73 participants’ retest responses at a two-week interval, 37 (48%) were excluded from the test–retest analyses because they had visited a general practitioner, medical specialist, or physiotherapist in the interim period. The test–retest subsample (*n* = 36) appeared broadly representative of the larger study cohort with respect to age, sex, language, employment, comorbidity, and most clinical variables. It contained proportionally more patients with prior joint replacement and somewhat higher educational attainment (Table 1).

The sociodemographic and clinical characteristics of the study population are shown in Table 1 and Table 2. The mean age was 71.9 years (SD 10.3), and 58.6% of the participants were female. Most patients reported German as their mother tongue (61.3%), followed by Italian (32.3%), Ladin (2.3%), and other languages (4.1%). Educational attainment varied, with 16.5% having completed only compulsory schooling, 22.9% vocational training, 37.6% secondary school, and 23.0% higher education or university. Most patients were retired (69.9%), with smaller proportions employed full-time (14.7%) or part-time (9.4%) jobs. Among patients with valid responses (*n* = 266), 35.3% rated their financial situation as very good or good, 52.6% as moderate, and 12.0% as poor or very poor. Clinically, 47.9% of the patients reported right knee involvement and 34.3% reported left knee involvement, 26.7% had previously undergone joint replacement surgery, and 61.5% reported at least one chronic comorbidity.

Patients most frequently reported their general practitioner (65.4%) and orthopedic surgeon (50.4%) as sources of information on OA, followed by physiotherapists (16.5%). Only a minority mentioned other sources, including rheumatologists (7.5%), the internet or media (5.3%), family or friends (4.9%), pharmacists (1.5%), nurses (0.8%), or other patients (0.8%). While 55.3% of patients reported using a single source of information, 27.4% used two sources, 10.9% used three sources, and only small proportions reported four (3.8%) or five (0.8%) different sources. Regarding the absence of information, 7.9% explicitly indicated that they had not received information from any source, yet only 1.9% had no source recorded at all, suggesting that some participants marked both “no source” and at least one source, resulting in inconsistent responses.

### 3.2. OA-QI Achievement Scores

At the item level, the proportion of patients who reported receiving recommended care varied widely across the 17 OA-QI indicators (Table 3). The highest achievement rates were observed for advice on physical activity (81.6%), NSAID prescription (80.3%), and assessment of joint pain (77.2%). In contrast, the lowest rates were recorded for discussions on occupational aspects (18.2%), support for weight reduction (33.5%), and assessment of the need for walking aids (31.0%).

At the patient level, the mean OA-QI achievement score was 58.7% (SD 28.5), with a median score of 60.8% (IQR 35.3–84.9). This distribution indicates that, on average, patients reported receiving slightly more than half of the recommended quality indicators, with marked variability across individuals.

Floor and ceiling analyses showed that only 1.5% of patients scored 0% and 8.3% achieved the maximum score of 100%. As both values were below the predefined 15% threshold, no floor or ceiling effects were observed.

### 3.3. Comprehensibility and Acceptability

Among the 38 patients who completed the comprehensibility and acceptability module, 94.7% rated all questions as “very understandable” or “rather understandable,” while 5.3% considered at least some items to be difficult. A total of 10.5% of the participants reported that individual questions were difficult to understand, and 18.4% noted unclear terms. Regarding ease of completion, 94.7% described the questionnaire as “very easy” or “rather easy,” whereas 5.3% found it difficult, and 10.5% felt that completing the questionnaire took too long. Finally, 86.8% of the participants considered the response categories to be clear and appropriate, whereas 13.2% indicated difficulties with them.

There were only three comments in the open-text fields. Two respondents mentioned that they found the German and Italian terms for “osteoarthritis” challenging to comprehend, while one patient pointed out that some questions seemed redundant. These remarks did not concern the OA-QI items but additional sociodemographic and clinical variables that had been included in the questionnaire.

### 3.4. Test–Retest Reliability

The test–retest reliability was evaluated among 36 participants who remained stable and did not consult a healthcare professional during the interim period. For the overall OA-QI achievement score, the intraclass correlation coefficient (ICC, two-way mixed effects, absolute agreement, single measures) was 0.55 (95% CI 0.28–0.74; *p* < 0.001), indicating moderate reliability. The corresponding Cronbach’s alpha for the two measurement occasions was 0.71. The standard error of measurement (SEM) was 20.7, resulting in the smallest detectable change (SDC) of 57.3 points at the individual level and 9.6 points at the group level (*n* = 36).

At the item level, κ values ranged from 0.33 to 0.69, with percent agreement between 61.1% and 86.1% (Table 4), indicating acceptable reliability. Most items demonstrated moderate reliability, with several reaching substantial reliability. Information items showed κ = 0.51 for information about osteoarthritis and κ = 0.53 for information about treatment options. Self-management advice also yielded substantial reliability (κ = 0.64). The item on receiving help for weight loss, which previously showed poor agreement, now had fair reliability (κ = 0.38). Items regarding pharmacological treatment and side effects demonstrated only fair to moderate agreement, with κ values of approximately 0.40–0.46. The newly added version 3 item on surgical referral achieved moderate reliability (κ = 0.55).

### 3.5. Associations with Additional Variables

At the bivariate level, OA-QI achievement scores were largely unrelated to age, sex, pain frequency, activity limitation, symptom duration, number of joints affected, or prior surgery. In contrast, several sociodemographic and contextual factors showed small but significant associations. Italian-speaking patients reported higher OA-QI scores than German speakers (ρ ≈ 0.13, *p* = 0.03), while vocational and university education were associated with lower scores than mid-level education (ρ = −0.17 to −0.24, *p* ≤ 0.02). Patients in full-time employment and those who rated their financial situation as poor also reported lower OA-QI scores (ρ ≈ −0.12 to −0.18, *p* ≤ 0.05). The presence of at least one chronic comorbidity was correlated with reduced OA-QI scores (ρ = −0.13, *p* = 0.045).

The strongest positive associations were observed for information behavior: both the number of information sources used (ρ = 0.34, *p* < 0.001) and categorized information use (none, one, ≥2 sources; ρ = 0.32, *p* < 0.001) correlated with higher OA-QI scores. Overall, these findings suggest that patient education, language, socioeconomic context, and access to information play a more important role in reported care quality than clinical severity. An overview of the correlation results is presented in Table 5.

#### Multivariable Mixed-Effects Linear Regression

Regression analyses were conducted to examine independent associations between patient characteristics and OA-QI achievement scores, while accounting for clustering at the practice level.

Results are shown in Table 6. Female patients had significantly lower OA-QI scores than male patients (β = −6.18, 95% CI −11.62 to −0.74; *p* = 0.026). Each higher level of education was associated with an average 2.65-point higher OA-QI score (95% CI 0.81–4.49; *p* = 0.005). Greater activity limitation was independently associated with lower OA-QI scores (β = −3.71, 95% CI −6.19 to −1.23; *p* = 0.004). No significant associations were observed between age, economic status, comorbidity, prior surgery, number of affected joints, pain frequency, symptom duration, or language group.

The intraclass correlation coefficient (ICC) indicated that approximately 6% of the variance in OA-QI scores was attributable to differences between GP practices. Collinearity diagnostics indicated no evidence of problematic multicollinearity, with all VIF values < 1.5 and tolerance values > 0.7. The model fit statistics indicated an adequate fit (−2LL = 2276.9, AIC = 2280.9, BIC = 2287.9).

## 4. Discussion

This feasibility study introduced the OA-QI questionnaire version 3 in German and Italian to South Tyrol general practice and confirmed its applicability and reliability in the healthcare context. The achievement of quality indicators varied, with high rates for physical activity advice, NSAID prescription, and joint pain assessment, but low attainment for weight reduction support, occupational aspects discussion, and walking aid needs assessment. The mean achievement score was 58.7%, indicating that slightly more than half of the recommended indicators were met. Test–retest analyses showed fair to substantial reliability at the item level (κ range 0.33–0.69, agreement 61–86%), with the new surgical referral item showing moderate stability. At the total-score level, reliability was moderate (ICC = 0.55, 95% CI 0.28–0.74), with a standard error of measurement of 20.7 and smallest detectable change of 57.3 points at the individual level (9.6 at the group level). These results indicate that while OA-QI v3 can detect small changes in group-level achievement scores (≈10 percentage points), changes in individual patients must exceed 57 percentage points to be distinguished from the measurement error.

The main objective was to determine whether OA-QI v3 could be implemented in routine general practice, whether it was acceptable to patients, and whether it demonstrated reliability in this cultural and organizational setting in South Tyrol. The achieved main sample (*n* = 266) met the planned target and provided adequate precision for feasibility outcomes. However, the test–retest subsample (*n* = 36) remained smaller than ideal for precise estimation of intraclass correlations, limiting the stability of the reliability coefficients. Recruitment of targeted general practitioners and patients demonstrated the practicality of integrating the instrument into daily workflows. Patient feedback confirmed high comprehensibility and ease of completion in German and Italian, indicating their suitability for bilingual regions. The patterns of missingness were low, although items such as occupational aspects and weight management were less frequently applicable, reflecting domain relevance in the local context. The test–retest findings at the item and total score levels support the instrument’s stability and reliability. These feasibility outcomes demonstrate that OA-QI v3 is practicable in South Tyrol and generates consistent results that can serve as a basis for broader applications and international benchmarking of OA care.

### 4.1. Feasibility, Reliability, and Comparison with Other OA-QI Versions

The findings of this study confirm that OA-QI v3 is feasible and reliable in South Tyrol, with high patient acceptability using German or Italian translations. These results extend the evidence base of the OA-QI, which has undergone a staged process of development, revision, and cross-cultural adaptation since its inception. The original Norwegian OA-QI demonstrated content validity and fair-to-substantial reliability (κ = 0.20–0.80, agreement = 62–90%) [43]. A revised version (OA-QI v2) further improved the measurement properties, showing excellent reliability at the total score level (ICC = 0.89), acceptable construct validity, and responsiveness suitable for evaluative purposes [25]. The present findings of moderate total score reliability and item-level stability are consistent with the literature, confirming their applicability in a new bilingual setting.

Several language versions have been developed internationally. The Dutch translation followed forward–backward procedures and pre-testing with patient representatives, resulting in an 18-item version with acceptable internal consistency (α = 0.79) and achievement patterns similar to those observed in other countries [37]. The OA-QI has also been applied in Portuguese and English in cross-country comparisons with consistent psychometric performance [44]. In the United Kingdom, a patient-reported OA-QI (UK) was developed with the strong involvement of a Research User Group as part of the MOSAICS study, harmonizing several items with the Norwegian version and aligning the instrument with the NICE quality standards [26]. These parallel developments highlight the adaptability of OA-QI instruments to local contexts while retaining their core international comparability.

The German and Italian versions were prepared through professional translation, followed by adaptation by native-speaking rheumatologists in accordance with the recommended cross-cultural adaptation procedures [37,44,45]. Acceptability and reliability were confirmed using patient feedback and retesting at the pooled level. Because analyses were conducted on the pooled bilingual sample, potential linguistic or cultural differences between the German and Italian versions could not be assessed. Given the limited number of Italian-speaking participants, these data should not be interpreted as evidence of full cross-language equivalence. Future studies with larger Italian-speaking samples are needed to confirm the measurement equivalence across languages.

### 4.2. Interpretation and Comparison with Previous Literature

Across international primary care systems, the results of this survey replicate a well-described pattern in OA quality indicator achievement: relatively high delivery of lifestyle/exercise advice and routine pharmacotherapy, contrasted with persistent shortfalls in weight management support, occupational guidance, and assessment for assistive devices. Concordant findings have been reported in multi-country surveys and systematic reviews from the United Kingdom, Netherlands, Canada, Australia, Denmark, Norway, Portugal, and Singapore, suggesting system-wide rather than region-specific implementation gaps [23,24,44,46,47,48,49,50]. Patient-reported assessments echo these discrepancies, with many individuals indicating insufficient information and support for non-pharmacological care [26,44,46,47].

While average achievement is suboptimal across settings, specific profiles vary, likely reflecting differences in local priorities, access to multidisciplinary services and system design [44,46,48]. The recurrent under-delivery of weight-loss support and functional/assistive assessments aligns with evidence that indicators fitting established workflows (exercise advice, NSAID prescribing) are easier to operationalize, whereas indicators requiring coordinated behavior change or multidisciplinary input (weight management, occupational advice, walking-aid assessment) face greater barriers [51,52,53,54,55,56,57].

Limited implementation of core non-pharmacologic indicators, such as weight reduction support and exercise or physiotherapy referral, has important functional implications. Evidence consistently shows that inadequate delivery of these interventions is associated with greater pain intensity, reduced mobility, and accelerated functional decline in people with OA [58,59]. Strengthening these process indicators in primary care could therefore contribute directly to preserving independence and quality of life.

Taken together, the international literature positions our results within a consistent cross-system profile of strengths in lifestyle and basic pharmacologic management and persistent gaps in weight, work/role, and assistive device domains. These gaps are plausibly driven by structural constraints (time, referral pathways, reimbursement, limited integration of physiotherapy/occupational therapy), as well as clinician and patient beliefs that dampen the uptake of recommended non-pharmacologic care [53,54,55,56]. This comparison underscores the need for targeted implementation strategies in South Tyrol, embedding multidisciplinary pathways, practical weight-management support, and consistent assessment of functional aids so that guideline priorities translate into routine primary care.

### 4.3. Implications for Clinical Practice and Quality Monitoring

The study demonstrates that the bilingual OA-QI is suitable for use in South Tyrolean general practice, offering a reliable and patient-friendly way to assess care quality and highlight differences across the domains of OA management. In daily practice, the instrument can help general practitioners systematically identify gaps and address them more effectively, such as in weight management, occupational aspects, and assistive device provision.

The OA-QI is particularly valuable for quality monitoring at the group level. While individual-level reliability is limited, requiring very large changes in scores to exceed measurement error, group-level reliability is strong and allows for the meaningful detection of smaller changes in average scores. This mirrors the findings from the revised Norwegian OA-QI v2, where group-level reliability was excellent, but the measurement error at the individual level was high [25]. Similar patterns have been reported for other patient-reported instruments, including the EuroQol 5-Dimension 5-Level questionnaire (EQ-5D-5L), Knee Injury and Osteoarthritis Outcome Score (KOOS), Hip Disability and Osteoarthritis Outcome Score (HOOS), Osteoarthritis Knee and Hip Quality of Life questionnaire (OAKHQOL), and Osteoarthritis Quality of Life scale (OAQoL). These measures consistently show high reliability and validity for group comparisons but limited responsiveness for tracking individual patients [60,61,62,63,64,65,66,67].

Aggregated OA-QI results can thus form a basis for practice-level quality work, regional training initiatives, and benchmarking across practices and internationally [25,26,37]. In this way, the instrument contributes to both transparency and quality-oriented improvement efforts.

For primary care professionals, the OA-QI v3 provides a feasible and patient-centered approach to monitor adherence to key osteoarthritis care processes. Its use can support practice-level benchmarking, stimulate team-based reflection on modifiable care gaps, and guide targeted quality improvement initiatives, particularly in non-pharmacologic management areas such as exercise, weight reduction, and patient education.

### 4.4. Strengths and Limitations

This study is the first to apply the OA-QI v3 in a bilingual healthcare setting, demonstrating its feasibility in South Tyrolean general practice. The recruitment rates were high, missing data were low, and patient feedback confirmed good comprehensibility. Feasibility studies typically do not require formal power calculations; however, guidance emphasizes stable estimates for recruitment, acceptability, and missingness [32,39]. Reliability was assessed using multiple metrics, and the findings replicated international patterns of OA care quality, with strengths in lifestyle advice and pharmacotherapy but gaps in weight management and functional support.

However, several limitations temper these strengths. The study was conducted in a motivated GP network (SAMNET), which may limit its generalizability. As this study was conceived as a feasibility pilot rather than a full validation, certain elements of the published protocol [28] could not be implemented, such as the language-specific reliability testing. Nevertheless, this process provided important lessons, including the risk of dropout in retest groups due to clinical instability.

An additional limitation pertains to the lack of a formal back-translation step and structured cognitive debriefing with both language groups during the translation process. Although the bilingual adaptation adhered to professional translation and expert review protocols, the study did not encompass full cross-cultural validation as recommended by Beaton et al. [45]. Consequently, future research should incorporate formal cross-cultural adaptation and measurement invariance testing to verify equivalence between the German and Italian versions of the OA-QI v3.

Given the small number of Italian-speaking participants (<10), subgroup analyses of acceptability and reliability would have been statistically underpowered.

The retest subgroup differed from the full cohort (e.g., more post-surgery patients and educational differences), and because approximately half had contact with a healthcare professional during the interval, indicating potential changes in their clinical status, the effective sample for reliability analyses was small. The relatively small retest subsample, which included a higher proportion of participants with prior joint replacement and higher educational attainment, limits the precision and generalizability of the reliability estimates. The exclusion of nearly half of the original retest group due to interim healthcare visits may also have introduced selection bias, potentially excluding less stable or more symptomatic patients. This underscores the need to account for patient stability when planning retest studies. Finally, reliance on retrospective self-reports carries the risk of recall bias.

The planned cross-national study in South Tyrol and Austria should follow the original protocol [28], ensure larger and clinically stable subgroups for test–retest analyses, and include cognitive debriefing in the study design. Broader psychometric testing, including responsiveness and construct validity, is needed to establish the OA-QI as a robust tool for quality monitoring and international benchmarking.

## 5. Conclusions

This feasibility pilot demonstrated that the German and Italian OA-QI v3 can be integrated into South Tyrolean general practice, with high patient acceptability and adequate reliability for group-level monitoring. Achievement rates revealed strengths in lifestyle advice, pharmacological management, and pain assessment but persistent shortfalls in weight reduction support, occupational aspects, and assistive device provision, mirroring international patterns. The study showed that the OA-QI is most reliable for evaluating care quality at the group level, whereas measurement error limits its use for monitoring individual patients. Lessons learned from the retest phase, particularly the influence of subgroup composition and clinical instability on reliability estimates, are directly relevant for future validation studies. The present results provide preliminary reliability estimates for the OA-QI v3 that should be confirmed in larger and more representative samples. Overall, the findings support the OA-QI as a practical and informative tool for identifying care gaps, guiding quality improvement, and enabling international benchmarking. Cross-national validation in Italy and Austria will be necessary to establish full psychometric evidence for the German and Italian versions. Future studies should replicate these findings in diverse linguistic and clinical contexts to further establish the external validity and transferability of the OA-QI v3 for international use in primary care.

## Figures and Tables

**Table 1 medicina-61-01921-t001:** Sociodemographic of the study sample (*n* = 266) and test–retest subsample (*n* = 36).

Sociodemographic Characteristics	*n* (%) or Mean (SD)	
	Study	Test–Retest
Age, years (mean, SD)	71.9 (10.3)	72.6 (10.0)
Age group		
<60 years	35 (13.2)	5 (13.9)
60–69 years	74 (27.8)	10 (27.8)
70–79 years	93 (35.0)	12 (33.3)
≥80 years	64 (24.1)	9 (25.0)
Sex		
Female	156 (58.6)	23 (63.9)
Male	103 (38.7)	12 (33.3)
Other/missing	7 (2.7)	1 (2.8)
Mother tongue		
German	163 (61.3)	21 (58.3)
Italian	86 (32.3)	11 (30.6)
Ladin	6 (2.3)	2 (5.6)
Other	11 (4.1)	2 (5.6)
Education		
Compulsory school	144 (54.1)	13 (36.1)
Vocational training	61 (22.9)	4 (11.1)
Highschool	43 (16.2)	16 (44.4)
University	18 (6.8)	3 (8.3)
Employment status		
Retired	186 (69.9)	24 (66.7)
Full-time	39 (14.7)	6 (16.7)
Part-time	25 (9.4)	3 (8.3)
Unemployed/Other	4 (1.5)	3 (8.3)
Subjective economic status		
Very good	12 (4.5)	2 (5.6)
Good	82 (30.8)	8 (22.2)
Moderate	140 (52.6)	25 (69.4)
Poor	30 (11.3)	1 (2.8)
Very poor	2 (0.8)	0

Abbreviation: SD, standard deviation.

**Table 2 medicina-61-01921-t002:** Self-reported clinical characteristics of the study sample (*n* = 266) and test–retest subsample (*n* = 36).

Clinical Characteristics	*n* (%)	
	Study	Test–Retest
Duration of OA symptoms		
<1 year	27 (10.3)	1 (2.8)
1–5 years	45 (17.2)	6 (16.7)
6–10 years	52 (19.8)	6 (16.7)
11–15 years	59 (22.5)	7 (19.4)
16–20 years	28 (10.7)	6 (16.7)
>20 years	22 (8.4)	4 (11.1)
Pain frequency		
Daily	143 (53.6)	15 (42.9)
Weekly	50 (18.8)	9 (25.7)
Rarely	59 (22.2)	9 (25.7)
Never	14 (5.4)	(5.7)
Activity limitation in daily life		
None	41 (15.5)	4 (11.1)
Mild	97 (36.6)	12 (33.3)
Moderate	102 (38.5)	15 (41.7)
Severe	25 (9.4)	5 (13.9)
Joints affected ^1^		
Left hip	60 (21.4)	7 (19.4)
Right hip	74 (26.4)	11 (30.6)
Left knee	96 (34.3)	15 (41.7)
Right knee	134 (47.9)	19 (52.8)
Prior joint replacement surgery ^2^	71 (26.7)	26 (96.3)
≥1 comorbidity ^3^	164 (61.5)	21 (60.0)
Physiotherapy (ever)	119 (45.1)	17 (47.2)
Any lifestyle change vs. none ^4^	141 (53.0)	14 (38.8)
Imaging (X-ray/MRI performed)	201 (75.5)	30 (83.3)

^1^ Percentages do not sum to 100% because multiple joints may be affected. ^2^ Percentages refer to patients with at least one joint replacement; multiple surgeries were possible. ^3^ Comorbidities were self-reported by the patients and were not validated against medical records. ^4^ Lifestyle changes refer to patient-reported changes in diet and/or physical activity following OA diagnosis.

**Table 3 medicina-61-01921-t003:** Item-level achievement of the 17 Osteoarthritis Quality Indicators (*n* = 266).

Item	Osteoarthritis Quality Indicators	Valid *n*	Yes, *n* (%)
1	Information about OA	259	172 (66.4)
2	Information about treatment options	261	177 (67.8)
3	Counseling on coping with OA in daily life	262	160 (61.1)
4	Advice on physical activity	257	210 (81.6)
5	Instruction in physical exercises	261	163 (62.5)
6	Advice on weight reduction (if overweight)	206	75 (36.4)
7	Support with weight reduction (if overweight)	179	60 (33.5)
8	Assessment of joint pain	254	196 (77.2)
9	Assessment of daily activity limitations	256	164 (64.1)
10	Assessment of need for walking aids	187	58 (31.0)
11	Discussion of occupational aspects	187	34 (18.2)
12	Follow-up/control visits	259	159 (61.4)
13	Prescription of NSAIDs	234	188 (80.3)
14	Information about NSAID side effects	214	118 (55.1)
15	Corticosteroid injection offered	216	129 (59.7)
16	Information about corticosteroid side effects	182	100 (54.9)
17	Information/referral for joint replacement	239	152 (63.6)

Abbreviation: NSAID, nonsteroidal anti-inflammatory drug.

**Table 4 medicina-61-01921-t004:** Test–retest reliability over a 2-week period for single items in the OA-QI v3-D or OA-QI v3-I questionnaire (*n* = 36).

Item	Quality Indicator Items	Kappa *	95% CI ^†^	Agreement (%) ^§^
1	Have you been offered information about osteoarthritis?	0.509	0.25–0.77	75.0
2	Have you been offered information about treatment options for your osteoarthritis?	0.525	0.23–0.82	80.6
3	Have you been offered information about how you can manage your osteoarthritis?	0.640	0.39–0.89	83.3
4	Have you been advised that exercise and physical activity are important to help your osteoarthritis?	0.579	0.25–0.91	86.1
5	Have you been offered guidance on how you can exercise your joints and be physically active?	0.497	0.21–0.78	75.0
6	If you are overweight, have you been advised to try losing weight?	0.460	0.23–0.69	63.9
7	Have you been offered or given help to lose weight?	0.382	0.14–0.63	61.1
8	Has a health professional discussed with you any problems you may have with daily activities due to your osteoarthritis?	0.344	0.05–0.64	72.2
9	If you have trouble walking, has someone discussed with you if a walking aid like walking sticks, cane, or crutch might be helpful?	0.496	0.23–0.75	72.2
10	If you have trouble working due to your osteoarthritis, have you been offered advice about how to remain in or return to paid or unpaid work?	0.692	0.49–0.89	80.6
11	Has a health professional asked you about your joint pain?	0.328	0.07–0.59	61.1
12	Has a health professional discussed with you when you should return for another consultation for your osteoarthritis?	0.418	0.16–0.68	66.7
13	Were non-steroidal anti-inflammatory medications the first medication that was recommended to you? (e.g., ibuprofen, diclofenac, naproxen, celecoxib)	0.397	0.10–0.70	75.0
14	If you use non-steroidal anti-inflammatory medication, have you received information about possible side effects?	0.435	0.18–0.69	66.7
15	If you are severely troubled by pain and other approaches do not help or are unsuitable, have you been offered a steroid injection?	0.464	0.23–0.70	66.7
16	If you were offered a steroid injection, were you offered information about possible side effects?	0.421	0.19–0.65	61.1
17	If you are severely troubled by your osteoarthritis and exercise, medication, or other approaches do not help, have you been offered a referral for an assessment for surgery? (e.g., joint replacement)	0.553	0.31–0.80	74.3 ^1^

Item response alternatives: items 1–5: ‘Yes’, ‘No’ or “Don’t remember”; items 6–7: ‘Yes’, ‘No’ or “Not overweight”; items 8–10: ‘Yes’, ‘No’ or “Not relevant”; item 11: ‘Yes’, ‘No’ or “Don’t remember”; item 12: ‘Yes’, ‘No’ or “Not relevant/unsuitable medication”; item 13: ‘Yes’, ‘No’ or “Not relevant”; item 14: ‘Yes’, ‘No’ or “Don’t remember”; item 15: ‘Yes’, ‘No’ or “Not severely troubled/Not relevant”; items 16–17: ‘Yes’, ‘No’ or “Not relevant”. * Cohen’s kappa statistic; 0.21–0.40 = fair, 0.41–0.60 = moderate, 0.61–0.80 = substantial, >0.80 = almost perfect agreement [42]. ^†^ CI, 95% confidence interval of kappa calculated using the asymptotic standard error. ^§^ The percent agreement represents the proportion of identical responses at baseline and retest among valid cases. ^1^
*n* = 35.

**Table 5 medicina-61-01921-t005:** Bivariate correlations between OA-QI achievement and patient characteristics.

Variable ^1^	Correlation (ρ or r) ^2^	*p*-Value	Direction	Effect Size ^3^
Age (years)	−0.05	0.37	n.s.	Negligible
Sex (female)	0.02	0.68	n.s.	Negligible
Language (Italian)	0.13	0.03	↑	Small
Language (Ladin/other)	−0.07	0.24	n.s.	Negligible
Education—vocational	−0.24	<0.01	↓	Small–moderate
Education—university	−0.15	0.02	↓	Small
Employment—full-time	−0.13	0.04	↓	Small
Economic status (worse)	−0.12 to −0.18	0.003–0.05	↓	Small
≥1 chronic disease	−0.13	0.045	↓	Small
Prior surgery	0.01	0.84	n.s.	Negligible
Number of affected joints	−0.06	0.31	n.s.	Negligible
Pain frequency	0.03	0.57	n.s.	Negligible
Activity limitation	−0.04	0.42	n.s.	Negligible
Duration of symptoms	−0.02	0.73	n.s.	Negligible
Number of information sources	0.34	<0.001	↑	Moderate
Information use (categorical)	0.32	<0.001	↑	Moderate

^1^ Variables were assessed as follows: age (years, continuous); sex (0 = male, 1 = female); language group (dummy variables with German as reference: Italian, Ladin, other); education (ordinal categories: compulsory school, vocational training, high school, university); employment status (dummy variables with retirement as reference: full-time, part-time, unemployed/other); economic status (five-point self-rating from very good to very difficult, treated as ordinal); comorbidities (0 = none, 1 = ≥1 chronic condition); prior surgery (0 = no hip/knee replacement, 1 = any replacement); number of affected joints (0–4: left/right hip, left/right knee); pain frequency (ordinal: never, rarely, weekly, daily); activity limitation (ordinal: no, sometimes, often, constantly); duration of symptoms (ordinal: <1 year to >20 years); and information sources (sum of up to 10 possible sources, and categorical: none, one, ≥2). ^2^ Spearman’s rank correlation coefficient ρ (Pearson’s r for age). ^3^ Effect sizes were interpreted following Cohen’s thresholds (negligible < 0.1, small ≈ 0.1, moderate ≈ 0.3, and large ≥ 0.5). Abbreviations: n.s., not significant. Arrows denote direction of correlation: ↑ positive, ↓ negative.

**Table 6 medicina-61-01921-t006:** Mixed-effects linear regression of OA-QI achievement scores, including patient- and practice-level predictors (random intercepts for general practitioners).

Predictor *	β (95% CI)	*p*-Value
Female sex	−6.18 (−11.62, −0.74)	0.026
Education (per higher category)	+2.65 (0.81, 4.49)	0.005
Activity limitation (per severity level)	−3.71 (−6.19, −1.23)	0.004
Age (years)	−0.01 (−0.13, 0.11)	0.845
Economic status (per category worse)	−0.91 (−2.08, 0.26)	0.129
Comorbidity (≥1 vs. none)	−2.07 (−6.84, 2.71)	0.395
Prior joint surgery (yes vs. no)	−0.12 (−5.36, 5.12)	0.964
Number of joints affected	−0.38 (−2.30, 1.55)	0.701
Pain frequency (per severity level)	−0.21 (−1.81, 1.39)	0.794
Duration of symptoms (years)	−0.06 (−0.87, 0.75)	0.892
Language: Italian vs. German	+0.54 (−4.27, 5.34)	0.823
Language: Ladin vs. German	−1.79 (−13.18, 9.61)	0.755
Language: Other vs. German	+4.62 (−7.41, 16.64)	0.448

* Education and economic status were modeled as ordinal variables. Activity limitations and pain frequency were coded according to severity. Binary predictors were coded 0/1 with reference categories of male sex, no comorbidity, and no prior joint replacement surgery. The German language group was used as a reference. CI: confidence interval.

## Data Availability

Data are available from the corresponding author upon reasonable request.

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
