# Peer review of "Feasibility and Reliability of the Osteoarthritis Quality Indicator Questionnaire for Assessing Osteoarthritis Care in Bilingual General Practices in South Tyrol/Alto Adige, Italy"

_medicina, 2025, doi:10.3390/medicina61111921_

Round 1
Reviewer 1 Report
Comments and Suggestions for Authors
1) Summary
This study evaluates the feasibility and test-retest reliability of the German and Italian versions of the Osteoarthritis Quality Indicator questionnaire version 3 (OA-QI v3) within the bilingual primary care context of South Tyrol, Italy. The research involved 38 general practitioners who recruited 266 patients with hip or knee osteoarthritis. Patients completed the OA-QI v3, with subsamples participating in comprehensibility testing (n=38) and a two-week retest for reliability (n=36). The study successfully demonstrated high recruitment rates and patient-reported acceptability. Reliability analysis showed fair-to-substantial agreement at the item level (κ = 0.33–0.69) and moderate reliability for the total score (ICC = 0.55), sufficient for group-level comparisons but not individual patient monitoring. The findings position the OA-QI v3 as a practical tool for benchmarking osteoarthritis care quality in this unique bilingual setting and similar primary care environments.
2) Overall Assessment
This manuscript makes a valuable contribution by adapting and testing a standardized quality assessment tool in an under-researched bilingual healthcare context. The core strength lies in its rigorous methodological approach within a established practice-based research network, successfully demonstrating the tool's feasibility. The principal concerns relate to the limited sample size for the reliability analysis, the handling of the bilingual validation, and some methodological choices in the translation process. Nevertheless, the study achieves its stated aims and provides a solid foundation for future validation work.
3) Major Strengths
-
Addressing a Clear Gap: The study effectively addresses the lack of validated, culturally adapted instruments for assessing OA care quality in the Italian primary care context, and specifically the bilingual South Tyrol region. The introduction clearly establishes this need, stating, "The bilingual...healthcare context of the region has constrained the capacity of general practice networks to systematically measure quality."
-
Strong Pragmatic Design: The use of the South Tyrolean General Practice Research Network (SAMNET) ensured high participation rates (95% of targeted GPs and patients), demonstrating excellent feasibility for integration into routine care. This real-world testing enhances the ecological validity of the findings.
-
Comprehensive Feasibility Assessment: The study goes beyond simple implementation to thoroughly assess comprehensibility and acceptability from the patient perspective. The finding that "94.7% rated all questions as 'very understandable' or 'rather understandable'" is a strong indicator of the tool's patient-facing suitability.
-
Transparent and Detailed Reporting: The manuscript provides a clear description of the OA-QI v3, its scoring, and the statistical analyses. The reporting of reliability using multiple metrics (κ, ICC, SEM, SDC) is particularly thorough and informative for readers assessing the tool's potential applications.
4) Major Concerns
-
Underpowered and Potentially Biased Test-Retest Sample: The effective sample for test-retest reliability was small (n=36), and it "contained proportionally more patients with prior joint replacement and somewhat higher educational attainment" than the main cohort. Furthermore, 48% of the original retest sample was excluded due to interim healthcare visits, which may systematically exclude less stable or more symptomatic patients.
-
Fix: The discussion should more explicitly acknowledge that the reliability estimates, particularly the ICC, may be unstable due to the small sample and potential selection bias. The conclusion should strongly emphasize that these are preliminary reliability estimates that require confirmation in a larger, more representative retest sample.
-
-
Lack of Formal Cross-Cultural Validation for the Bilingual Context: The translation relied on a professional service and review by rheumatologists, but the absence of a formal back-translation step and cognitive debriefing with patients from both language groups during the adaptation phase is a notable methodological limitation, as per established guidelines like those of Beaton et al. (2000), which is cited.
-
Fix: This should be explicitly stated as a limitation in the discussion. The authors should clarify that this study focused on feasibility and reliability and that a full cross-cultural validation, including measurement invariance testing between the German and Italian versions, is a necessary next step.
-
-
Pooled Analysis Masks Potential Linguistic Differences: The analyses for comprehensibility and reliability were conducted on the pooled sample. With only 32.3% of the main cohort being Italian-speaking, the study is likely underpowered to detect meaningful differences between the language versions.
-
Fix: The discussion should more strongly caution against assuming full equivalence between the two language versions based on the current data. The statement that "subgroup analyses of acceptability and reliability would have been statistically underpowered" should be moved from the discussion (section 4.1) to the limitations section for greater emphasis.
-
5) Minor Comments
-
Abstract: The abstract ends abruptly with "providing important". Please check the final submitted version to ensure the sentence is complete.
-
Table 1 Formatting: The "Clinical Characteristics" section of Table 1 is difficult to parse. The layout is confusing, making it hard to quickly compare the study sample and test-retest subsample. Consider splitting this into two separate tables or significantly reformatting for clarity.
-
Methodology Description: In Section 2.4.1, the description of the transition from OA-QI v2 to v3 is slightly confusing. A concise table in the supplementary materials summarizing the key changes (e.g., specific items reworded, added, or removed) would be very helpful for readers familiar with previous versions.
-
Statistical Reporting: In the results for the multivariable regression (Table 5), it would be helpful to also report the results for the "number of information sources" variable, given it had one of the strongest bivariate correlations (r=0.34). Its absence from the multivariate model should be justified.
-
Clarity in Limitations: The limitation regarding the "small number of Italian-speaking participants" is currently buried in the discussion (4.1). It would be more transparent to include it in the dedicated "Strengths and Limitations" section (4.4).
-
AI Use Statement: The acknowledgment of AI use is commendably transparent. To further strengthen this, consider specifying which parts of the Methods section were refined with AI assistance.
6) Methodology & Rigor Check
-
The cross-sectional survey design is appropriate for the stated objectives of assessing feasibility and reliability.
-
The sample size for the main survey (n=266) is adequate and aligns with subject-to-item ratio recommendations.
-
Patient recruitment criteria (NICE-based) are clear and clinically relevant.
-
The handling of "not relevant" responses and the calculation of the achievement score follow established procedures from prior OA-QI studies, ensuring comparability.
-
The use of a 14-day retest interval is standard and appropriate for assessing reliability.
-
The exclusion of patients who had clinical consultations during the retest period is methodologically sound for assessing stability, but as noted, it impacts the final sample size and representativeness.
-
The statistical analyses for reliability (percent agreement, κ, ICC, SEM, SDC) are comprehensive and correctly applied.
7) Interpretation & Positioning
-
The conclusions are well-supported by the data. The authors correctly state that the tool is reliable for group-level assessment but not for monitoring individual patients, based on the high SDCindividual.
-
The discussion effectively positions the findings within the international context of OA-QI studies, correctly identifying consistent patterns of care (e.g., high exercise advice, low weight management support) across different healthcare systems.
-
The authors appropriately note that their GP network (SAMNET) may consist of more motivated practitioners, which could limit generalizability, a threat to external validity.
-
The implications for clinical practice and quality monitoring are realistic and well-argued, focusing on practice-level benchmarking rather than individual patient tracking.
8) Writing & Presentation
-
The manuscript is generally well-written, with a clear and logical flow.
-
The title and abstract accurately reflect the study's content and findings.
-
The figures and tables are relevant, though Table 1 requires improvement in presentation as noted.
-
Sample rewrite for improved clarity: In the Discussion, the sentence: "These gaps are plausibly driven by structural constraints (time, referral pathways, reimbursement, limited integration of physiotherapy/occupational therapy), as well as clinician and patient beliefs that dampen the uptake of recommended non-pharmacologic care" is excellent and clear. No major rewrites are needed.
9) Ethical & Compliance Notes
-
The study appears ethically sound, with appropriate IRB approval and informed consent documented.
-
The conflict of interest statement is clear.
-
The data availability statement is standard.
-
The funding source is transparently declared.
-
The acknowledgment of AI use is a model of good practice.
10) Actionable Revision Plan
-
Address Test-Retest Sample Limitation: Strengthen the discussion and limitations sections to explicitly state that the reliability estimates are preliminary due to the small and potentially selected retest sample. Emphasize the need for confirmation in a larger, prospectively defined stable cohort.
-
Clarify Validation Scope: Explicitly differentiate in the discussion between a feasibility/reliability study and a full cross-cultural validation. Acknowledge that the absence of back-translation and cognitive debriefing are limitations of the current adaptation process.
-
Reorganize Limitations: Move the discussion about the underpowered Italian subgroup analysis from section 4.1 to the formal "Strengths and Limitations" section (4.4) for greater prominence and transparency.
-
Improve Table 1: Redesign Table 1 to clearly and separately present the sociodemographic and clinical characteristics of the main sample and the test-retest subsample. This is a high-impact change for readability.
-
Justify Multivariate Model: In the results or methods section, briefly explain the rationale for the variable selection in the final multivariate model (e.g., why "number of information sources" was not included despite a strong bivariate correlation).
-
Final Proofread: Conduct a final proofread to ensure the abstract is complete and that all cross-references (e.g., to supplementary tables) are correct.
Reviewer 2 Report
Comments and Suggestions for Authors
Dear authors,
I appreciate the opportunity to review your manuscript, which addresses a specific and underexplored need: the assessment of quality of care in osteoarthritis using an instrument validated in a real-life bilingual primary care setting.
Please allow me to offer a few comments:
1. Title: Please consider including the full name of the instrument (avoiding unexplained abbreviations) and adding "Validation Study" to clearly indicate the type of study.
2. Abstract: It would also be helpful to include the methodological design of your research in the methods section.
3. Introduction: I believe you provide a timely contextualization of the problem. You could add conceptual frameworks for quality improvement in primary care, such as the Donabedian or Chronic Care Model.
Likewise, it may be of interest if you could link the limited implementation of indicators with the functional consequences in people with OA.
4. Methodology: I believe the necessary sample size needs to be justified.
5. Results: The 50% participant exclusion rate in the test-retest phase seems very high. I believe it may require further discussion, as it could reduce power and generate bias.
I suggest reconsidering the current format of the combined tables of sociodemographic and clinical characteristics (Table 1). The joint presentation in parallel columns, without clear separation, may hinder fluent reading and comparative interpretation.
6. Discussion: I find the link to practical implications for primary care professionals lacking.
7. Conclusions: I consider them adequate and moderate. I could add the advisability of replication in other linguistic and clinical contexts to strengthen the instrument's external validity.
Round 2
Reviewer 1 Report
Comments and Suggestions for Authors
Accept in present form
Comments on the Quality of English LanguageAccept in present form
Reviewer 2 Report
Comments and Suggestions for Authors
I thank the authors for their kind consideration of our comments.